# Using DNA Metabarcoding to Identify Floral Visitation by Pollinators



**Abigail Lowe** [1,2,*] **, Laura Jones** [1] **, Lucy Witter** [1,3] **, Simon Creer** [2] **and Natasha de Vere** [4]

1   Science Department, National Botanic Garden of Wales, Llanarthne SA32 8HG, UK;
    laura.jones@gardenofwales.org.uk (L.J.); lucy.witter@gardenofwales.org.uk (L.W.)
2   Molecular Ecology and Evolution Group, School of Natural Sciences, Bangor University,
    Bangor LL57 2UW, UK; s.creer@bangor.ac.uk
3   Institute of Biological, Environmental and Rural Sciences,
    Aberystwyth University, Aberystwyth SY23 3FL, UK
4   Natural History Museum of Denmark, University of Copenhagen, 1350 Copenhagen K, Denmark;
    natasha.de.vere@snm.ku.dk
*   Correspondence: abigail.lowe@gardenofwales.org.uk

**Abstract:** The identification of floral visitation by pollinators provides an opportunity to improve our understanding of the fine-scale ecological interactions between plants and pollinators, contributing to biodiversity conservation and promoting ecosystem health. In this review, we outline the various methods which can be used to identify floral visitation, including plant-focused and insect-focused methods. We reviewed the literature covering the ways in which DNA metabarcoding has been used to answer ecological questions relating to plant use by pollinators and discuss the findings of this research. We present detailed methodological considerations for each step of the metabarcoding workflow, from sampling through to amplification, and finally bioinformatic analysis. Detailed guidance is provided to researchers for utilisation of these techniques, emphasising the importance of standardisation of methods and improving the reliability of results. Future opportunities and directions of using molecular methods to analyse plant–pollinator interactions are then discussed.

**Keywords:** DNA metabarcoding; pollen; pollinators; pollen metabarcoding; plant–pollinator interactions; DNA barcoding; honeybees; bumblebees; hoverflies

## 1. Background

Understanding the relationship between plants and pollinators is vital for biodiversity conservation, food security, and ecosystem sustainability [1,2]. Worldwide, there are approximately 350,000 animal pollinator species, of which insects contribute a significant proportion [3]. Despite the importance of pollinators, evidence of declines in species richness and abundance are increasing across the globe [4,5]. The most significant drivers of decline are land use change, pesticides, climate change, pests, and pathogens [6–9].

DNA metabarcoding provides a powerful tool for investigating pollinator foraging preferences and should be a standard part of the ecologist's toolkit. The aim of this review is to describe the range of approaches and methods available, along with their opportunities and challenges. We thoroughly explore the ecological questions that can be answered from identifying floral visitation across a range of species and habitats and present a summary of findings from the literature. The entire pollen metabarcoding workflow is described along with considerations and guidance for each step, in the hope of inspiring more researchers to adopt these techniques.

Identifying floral visitation can provide an insight into the resources used by insects and the pollination services they deliver [10]. Whilst the methods described here do not directly detect the process of pollination [11], we use the term pollinators as a general term to refer to flower-visiting insects.

## 2. Methods for Identifying Floral Visitation by Pollinators

Floral visitation studies may be plant- or insect-focused. Examples of insect-focused methods include observational methods such as mark recapture using paint, plastic tags [12], or harmonic radar [13]. In addition, waggle dances, used by honeybees to communicate the location of resources to the colony [14], can be de-coded to elucidate forage preferences and behaviour [15]. Floral visitation may also be investigated by identifying the pollen collected by the insect. Pollen microscopy has been widely utilised for diet characterisation by identifying pollen grains obtained from body parts of individuals, e.g., mouthparts [16], scopa [17] and entire bodies [18,19], or honey [20,21] and nest provisions [22,23]. However, the identification of pollen grains to species level using light microscopy is difficult and time-consuming [24]. In recent years, automated machine learning systems have been developed to identify pollen from images and are showing great promise [25–27].

Pollen may also be identified by DNA metabarcoding: a process involving large-scale identification of unknown taxa within a mixed sample using DNA barcode markers and high-throughput sequencing [28–30]. The DNA contained in the sample is compared to a reference library composed of DNA sequences of a standard genetic marker. For plants, parts of the genes coding for ribulose bisphosphate carboxylase large subunit (*rbcL*) and maturase K (*matK*) are recommended as standard markers due to their universality across land plants [31]. However, the length of *matK* (around 800 bp) and the requirement for multiple primer combinations to gain taxonomic coverage makes it less suitable for amplicon-based metabarcoding [32]. Instead, additional markers such as the non-coding nuclear internal transcribed region ITS2, the *trnL* intron, and the non-coding intergenic spacer *trnH-psbA* are often used, either alone, or alongside *rbcL* for increased species discrimination [33]. DNA metabarcoding has been used to successfully identify pollen from provisions within nests [34–36], honey [37–39], proboscises [16,40], guts [41,42], and the legs or bodies of insects [43–45] (Table S1).

Shotgun metagenomics is an alternative tool which can be used to identify taxonomic diversity within a mixed sample using untargeted sequencing of genomic fragments mapped to whole genomes or barcode regions [46,47]. By mapping genome-skims to a constructed reference library of plastid genomes, Lang et al. [48] demonstrated quantitative identification of >97% taxa in mixed pollen samples. The advantages of metagenomic methods are the option of PCR-free processes which reduce possible amplification biases, the ability to output long read lengths, and the increased taxonomic resolution compared to targeted sequencing of specific regions [46,49]. The main limitation facing whole-genome studies is that currently, few whole plant genomes are available, resulting in difficulties assembling reference material [46]. A further promising approach is the use of reverse metagenomics to map long reads produced by the MinION to genomic skims, a method which has produced semi-quantitative identification of plant species in mixed pollen loads [49].

Plant-focused methods of identifying floral visitation provide an alternative perspective to insect-focused methods. Interactions between plants and pollinators can be characterised through observing which insects visit plants (plant-focused) [50–53]. Two methods are commonly adopted: timed observations of plants with the frequency of each insect visit recorded [53,54], and transect or plot walks where individuals within a survey area are identified when visiting plants [52,55,56]. For both methods, insects are either identified in the field or captured for later identification. An example of a more novel plant-focused approach to elucidating floral visitation is through the method of obtaining residual insect DNA from flowers [57]. Similarly, the identification of 'microbial signatures' specific to pollinators within nectar can also be used to elucidate visitation [58,59].

## 3. Plant vs. Pollinator Perspective of Foraging

Recording floral visitation from the perspective of the plant or the insect will yield varying information [60–63], and each method of recording visits from either perspective

has its advantages and disadvantages. Plant-focused surveys using visual observations are the most common method of analysing plant–pollinator interactions, providing a quantitative measure of the frequency of interactions between species [55]. A key advantage of visual surveys is that there is an opportunity to supplement observational data with environmental metadata such as the time of interaction [64], weather conditions [65], plant colour [66], and horticultural variety [67], which can be used to explore further questions surrounding foraging behaviour. In addition, the type of resource (pollen, nectar, or resin) collected by pollinators can be identified [55], a vital component of pollinator ecology. It is often possible to identify both the plants and insects to species, providing pollinators are retained for identification through morphology or DNA barcoding [68].

The characterisation of interactions between plants and pollinators using plant-focused observations are usually grouped at the species level [52] due to difficulties tracking individuals [12]. This means that quantitative data (e.g., frequency of visits) can only be gained at the species level and information regarding individual foraging trips is inaccessible. Moreover, the period of observation is often limited both spatially and temporally, resulting in a bias towards abundant pollinator species [63]. As a result, interactions may be missed [62] and those of rare individuals may appear more specialised than in reality [63]. As a result, sampling effort is a major determining factor of the number of relationships which are recorded [69]. Further, the method used to observe interactions (e.g., transects, timed observations) will also lead to biases which should be considered when constructing networks [70]. Increasing the sampling effort by increasing the time spent surveying can increase the likelihood of capturing rare interactions and thus reducing the incidence of specialisation [71]. Identifying floral visitation through molecular analysis of remnant DNA on flowers provides an opportunity to increase the temporal scope of plant-focused surveys, whilst increasing the likelihood of detecting rare interactions in comparison to plant-focused visit surveys [57].

The use of visual and electronic aids to track insects such as paint, plastic tags [12], or harmonic radar [13] provides information on individual foraging to be determined, offering a different perspective compared to plant-focused observations. DNA metabarcoding and pollen microscopy allow for an increased insight into interactions which may be missed through observations [42,61,72–75]. These methods are free from the spatial limitations of observations which come as a result of visual bias, e.g., height [19], as they provide a record of any resources which have been accessed by the individual which may be up to several kilometres away [76]. For example, by analysing pollen loads of bumblebees, Carvell et al. [77] found that the dominant plant in pollen loads was not always the plant the bee had been caught on, demonstrating that observation of floral networks does not reveal all interactions with visitors.

Arstingstall et al. [78] found that when comparing plant–pollinator networks characterised by DNA metabarcoding of pollen to those constructed from observations of foraging bees, networks constructed from molecular analysis had increased species richness and reduced specialisation. By identifying the pollen assemblage carried by insects, it is possible to gain a semi-quantitative measure of frequency of use per individual (discussed in detail within the methodological considerations) [44,79]. The collection of insects for pollen analysis also allows specimens to be retained for identification through traditional morphology or DNA barcoding [68].

Nevertheless, insect-focused methods of identifying floral visitation are not free from biases or limitations. Some insect-focused methods of tracking pollinators can also suffer from spatial limitations such as tag ranges [13]. During observations the time spent foraging can be recorded; however, it is difficult to distinguish the temporal range of pollen found on an insect's body. Further, the identification of pollen from insects does not provide information on whether plants were visited to collect pollen, or incidental pollen transfer through visitation for nectar or resin collection, or, indeed, pollen that has collected on the body of an insect whilst it has been flying. Interactions observed through visual surveys can be undetected using DNA metabarcoding and pollen microscopy, owing to their rarity [74],

pollen accessibility [73], or use for nectar with limited or no pollen production [42,80]. These factors reduce the amount of pollen transferred to the insect and therefore identified. However, whilst rare interactions may still be missed through the identification of pollen, they are more likely to be captured than through plant-focused surveys [63].

Whilst both pollen microscopy and DNA metabarcoding yield valuable individual-level information on foraging, identification of plant taxa using DNA eliminates the need for expert palynologists for microscopy. Although also time-consuming and initially expensive [81], molecular processes may be easily scaled up [82,83]. In pollen microscopy, a small sub-sample is fully identified and used to estimate the composition of the total pollen load [63], whereas molecular analysis can sample the entire pollen assemblage on the body of an insect [45,84]. Although there is some congruence between the taxa which are difficult to identify using microscopy and those which are indistinguishable using DNA, e.g., some taxa within the Rosaceae family [38], both methods may detect additional taxa when compared to the other [83,85]. In comparing pollen microscopy and DNA metabarcoding, several authors have found higher taxonomic resolution of plant taxa identified [16,86] and a greater number of species detected [83,85–87] using DNA metabarcoding. For example, when comparing the use of metabarcoding and microscopy to characterise pollen transport networks in moths, Macgregor et al. [16] found that metabarcoding detected more interactions per moth species. This was likely due to the increased discriminatory power of metabarcoding which allows some pollen types to be separated to a lower taxonomic level than through microscopy [16]. Both methods, however, are subject to the stochasticity of detecting rare taxa [87,88].

The method used to identify floral visitation is dependent on the type of question being asked. In order to create highly resolved plant–pollinator interaction networks, it is recommended that a combination of plant- and insect-focused methods are used [16,62,63,89].

## 4. Using DNA Metabarcoding to Answer Questions about Pollinator Foraging Preferences

The use of DNA metabarcoding to answer ecological questions about pollinator foraging preferences has increased rapidly over recent years alongside key methodological developments (Table S1). A range of taxonomic groups have been studied; however, the research is predominately focused on wild bees (e.g., *Bombus, Megachile, Osmia*), managed bees (e.g., *Apis mellifera, Tetragonula carbonaria*), and hoverflies (Syrphidae). The questions addressed can be broadly grouped into four topics: (1) How does foraging change throughout time and space? (2) How is foraging affected by resource availability? (3) How are resources partitioned between species and individuals in a plant–pollinator network? (4) What is the relationship between plant use and pollinator health?

### 4.1. How Does Foraging Change throughout Time and Space?

DNA metabarcoding provides a useful method for monitoring plant use across wide spatiotemporal scales, such as multiple countries or regions [90] and, when compared with historical data, time periods over decades or centuries [38,91,92]. The reproducibility of DNA metabarcoding allows for continued sampling of foraging across a species' entire flight period, providing an understanding of plant selection at specific time points. When assessing foraging habits of pollinators throughout the year, it is often found that the amount and diversity of pollen collected is strongly influenced by season, most likely influenced by the phenology of surrounding plants [45,93–95]. In addition to tracking contemporary foraging habits, DNA metabarcoding has been shown to be a useful tool for analysing pollen from historical specimens [91,96,97]. By sequencing plant DNA from pollen obtained from museum specimens, Simanonok et al. [97] successfully identified the plants used by an endangered bumblebee species over 100 years, vastly improving current knowledge of resource use and mechanisms of decline. Similarly, analysing the pollen DNA within UK honey and comparing the plant diversity to samples characterised 65 years prior using microscopy revealed landscape-scale shifts in foraging habits due to changes in agricultural intensification, crop use, and the spread of invasive species [38].

Long-range movements can be tracked by identifying pollen on migrating insects [40,98]. Suchan et al. [98] detected plant species endemic to Africa on butterflies using DNA metabarcoding, significantly improving the understanding of migration patterns which were previously limited when using traditional techniques. As well as increasing the spatial scale of studies, pollen metabarcoding has highlighted the importance of trees and woody species to pollinators, plants with flowers which are often visually restricted and therefore may be missed during observational surveys [37,99]. Whilst most of these spatial assessments of foraging focus on geographic differences, only one study has specifically demonstrated the ability of pollen metabarcoding to elucidate changes in resource use across elevational gradients to better understand how climatic changes in the environment impact foraging of a solitary bee [41].

*4.2. How Is Foraging Affected by Resource Availability?*

A key area of research in pollinator foraging ecology is understanding why specific plants are used and whether this is driven by preferences relating to characteristics of the plant, e.g., nectar quality [100], or simply a result of resource availability [101]. By conducting floral surveys and comparing the flowering plants available to the plants identified in honey using DNA metabarcoding, de Vere et al. [37] found that honeybees only used 11% of genera available. Park and Nieh [94] also used a metabarcoding method along with herbarium records to illustrate that honeybees used between 2.7 and 10% of flowering species available over three seasons.

Insect visitation can be influenced by the abundance of floral resources in a landscape [102], which is affected both temporally by plant phenology [103] and spatially by habitat type [104]. Timberlake [105] utilised a null model method and DNA metabarcoding of pollen samples collected from bumblebees within farmland to illustrate that floral choice was not always driven by the abundance of plant species, nor their nectar availability. By identifying plants which are visited more than expected compared to their abundance, management recommendations can be given for maintaining appropriate floral provision aimed at the effective conservation of bumblebees on farmland [105]. Likewise, Jones [106] found no significant correlation between the abundance of plant taxa in the landscape and the abundance of plants found in honey samples each month. However, Nürnberger et al. [107] found that the number of plant genera in pollen loads of honeybees identified by metabarcoding was lower when floral availability was reduced. Recent work by Quinlan et al. [108] suggests that whilst honeybees may sometimes preferentially select plants found in high abundance, this is dependent on the time of year and nutritional demand.

DNA metabarcoding can be used to monitor how spatiotemporal changes in resource availability across landscapes affect the diet of wild and managed bees [90,104,109–111]. By assessing honeybee diet across gradients of land use, multiple authors have found that the richness and diversity of pollen collected is not strongly linked to the composition of surrounding landscapes [39,83,111,112]. Instead, seasonality of resources appears to be the greatest driver of diet, irrespective of land use [93,95].

*4.3. How Are Resources Partitioned between Species and Individuals in Plant–Pollinator Networks?*

The use of DNA-based methods for identifying species interactions allows complex networks to be constructed and analysed [16,89]. Constructing accurate networks is important to help fully understand their structure, as the level of specialisation and generalisation of networks, species, or individuals can affect their robustness against environmental change [113,114].

A number of authors have used molecular approaches to assess resource partitioning within large plant–pollinator networks [16,84,115]. Elliott et al. [116] used DNA metabarcoding to construct an interaction network between honeybees, native bees, and the floral resources used to identify resource overlap. The number of known floral hosts of many species were increased compared to the previous literature based on observational studies, improving the understanding of how wild and introduced bees co-exist in a landscape [116].

The ability to identify an individual's entire pollen assemblage results in the valuable characterisation of interactions at varying hierarchical levels throughout a plant–pollinator community [117]. To date, of the studies that have identified resource partitioning within plant–pollinator networks using DNA metabarcoding, all have found that generalised networks or species are made up of specialised individuals [44,84,115,118]. This presents a promising area of research to further investigate the levels of specialisation and generalisation exhibited by pollinators.

*4.4. What Is the Relationship between Plant Use and Pollinator Health?*

Floral resources vary in the quality of their nectar and pollen rewards [100], and consequently, the diversity of resources used has been found to impact pollinator fitness [119]. Insights into the nutritional ecology of pollinators can be unearthed using DNA metabarcoding, by quantifying the relationship between plant taxa found and either the protein, carbohydrate, lipid, and amino acid content of pollen [111,120,121] or the physiological glycogen, lipid, and protein levels of insects themselves [122].

As well as affecting the nutritional quality of provisions, the plant species visited by pollinators may also influence the bacteria present in the nest [123]. DNA metabarcoding allows plant–microbe relationships to be explored, increasing the understanding of plant–pollinator interactions throughout an insect's lifecycle. The relationship between the diversity of pollen species collected and the diversity of the microbiome appears complex. However, both positive and negative associations have been found between particular pollen types and bacteria [124–126]. For example, Voulgari-Kokota et al. [126] found that the presence of Acinetobacteria in pollen provisions of solitary bees was positively associated with the presence of some taxa such as European goldenrod (*Solidago virgaurea*), oxeye daisy (*Leucanthemum vulgare*), and yarrow (*Achillea millefolium*), but negatively associated with spear thistle (*Cirsium vulgare*), red poppy (*Papaver rhoeas*), and sycamore (*Acer pseudoplatanus*).

The identification of pollen in nests has also been used to investigate the relationship between mass-flowering crops and the prevalence of parasites in nests of mason bees (*Osmia* spp.), finding that increased abundance of resources may help to reduce transmission by diluting parasite transmission through reducing visitation frequency per flower [127].

**5. Key Methodological Considerations for Using DNA Approaches and Their Challenges**

*5.1. Study Design and Sampling*

Careful considerations are required for every stage of the molecular approach, from the initial stages of study design to the downstream bioinformatic analysis (Table 1). Firstly, the nature of the study system must be considered in order to understand the information which will be produced. For example, sampling pollen from a single bee which is actively foraging will yield different results to pollen collected through pollen traps or honey, as the latter methods represent the foraging efforts of multiple bees over numerous trips [37]. In addition, morphological features such as body size and pilosity (hairiness) of insects can influence the number and diversity of pollen retained [128]. Pollen may be transferred from plants visited solely for nectar [55], and some plants do not produce nectar at all [129]. In addition, nectar can itself be contaminated with pollen as a result of plant visitors [130]. Therefore, molecular analysis of pollen generates information on which plants have been visited for both pollen and nectar collection. Another important consideration is that the presence of pollen on insects does not assume pollination has occurred [10], and therefore the identification of pollen represents floral visitation only. It is also important to consider that when identifying plant material within nest provisions, contamination may occur from multiple sources of plant DNA such as pollen provisions or leaf or soil material used to build nests [86].

Capturing methods such as on transect walks or during observations will also influence the number and diversity of insects caught and therefore the resulting sampling universe. The flight times of insects and phenology of plants must also be considered due to their influence on foraging. For example, sampling one species across its entire flight period

will provide a more complete picture of resources used compared to studies undertaken within a shorter time period, which have limited information on the total resources used. Further, the time of day at which pollinators are sampled will affect the resultant species collected [131].

The nature of pollen sampling from insect bodies results in a risk of cross-contamination occurring in the field; therefore, samples should be collected using a combination of nets and sterile tubes, with nets changed regularly and sterilised between surveys [44]. Airborne pollen may also contaminate samples [61], leading some authors to use thresholds to exclude rare taxa (reviewed in [132]) or removing all wind-pollinated species from analysis [133]. However, it should be noted that rare taxa may include real interactions, and some pollinators are known to visit wind-pollinated plants [134,135]. Further work to quantify the prevalence of residual pollen left on plants by insect visitors would be useful to infer thresholds for removal [78].

The method of preserving samples may also affect the success of the study [136]. Whilst successful sequencing of pollen from historical specimens is possible [91], samples should be preserved quickly to avoid degradation of DNA. Most pollen metabarcoding studies have preserved samples by freezing at −20 °C; however, recent work by Quaresma et al. [137] suggests that the use of silica gel for preserving pollen should not be overlooked, particularly when samples are collected by citizen scientists.

**Table 1.** Key considerations required for each step of the pollen metabarcoding workflow.

| Step | Description of Method | Consideration | Recommendations |
|---|---|---|---|
| Sampling | Plant DNA can be captured through a number of sampling methods: | Source of pollen influences information obtained | Collect insects in sterile pots and replace nets if any pollen transfer is suspected |
| | | Morphological features of insects, such as body size and pilosity (hairiness), can influence the amount of pollen retained | |
| | 1. Pollen obtained from individuals collected from light traps, on transects, or within observational plots | Capture methods influence the number and diversity of insects caught | |
| | 2. Pollen obtained from within nest provisions | Contamination may occur | |
| | 3. Pollen obtained from honey samples | Sampling period limits the knowledge which can be gained | |
| Sample preservation | Avoidance of DNA degradation | Preservation method may affect downstream success | Store pollen samples at −20 °C or dry using silica gel to limit degradation of DNA |
| DNA extraction | Extraction of DNA from pollen | Quantity of DNA obtained is affected by extraction method | Membrane-based commercial kits offer a fast and simple way of yielding DNA, although they are costly |
| | | Success of DNA extraction may depend on pollen type and source | Additional purification step is required for honey samples, e.g., Zymo OneStep PCR Inhibitor Removal Kit |
| | | Contamination may occur | Stringent cleaning procedures are required using 10% bleach solution before and after each process Use of filter tips Use of negative controls |
| Amplification | PCR amplification of extracted DNA using primers which target specific region of interest | Choice of marker will influence which taxa are recovered and their taxonomic resolution | We recommend a multi-locus approach using *rbcL* [138,139] and ITS2 [140,141] Primer recommendations in Table S2, Supporting Information |
| | | Contamination may occur | Stringent cleaning procedures are required using 10% bleach solution before and after each process Use of filter tips Use of positive and negative controls |
| | | Biases may be introduced through primer specificity | Complete three rounds of PCR per sample and pool |

**Table 1.** *Cont.*

| Step | Description of Method | Consideration | Recommendations |
|---|---|---|---|
| Multiplexing and library preparation | Addition of nucleotide sequences to primers to allow for pooling of samples and compatibility with sequencing platforms | Each method has a trade-off between multiple factors including overall cost, risk of contamination and PCR efficiency Tag-jumping can occur causing misidentification | Index strategy used should be based on research question and experimental set-up A two-step PCR approach allows for cost-effective indexing |
| Sequencing | Identification of nucleotide sequences | Sequencing strategy is dependent on choice of marker | Illumina MiSeq (2 × 300 bp) allows sequencing of *rbcL* and ITS2 |
| Reference library | Comparison of DNA sequences to a reference library for identification | Identifications made through DNA metabarcoding will only be as good as the reference library | Create a reference library which is appropriate to the question being asked and ensure that it is complete and well curated |
| Bioinformatic analysis | Automated processes used to curate sequences for analysis including quality control | Species may be incorrectly assigned during automated processes | Requires manual verification steps by someone with knowledge of relevant plant taxa |
| | | Metabarcoding data are considered to be semi-quantitative | Treat proportion of sequences as relative read abundance for analysis |

## 5.2. DNA Extraction

Numerous DNA isolation methods exist which can influence the quality of the DNA template [142,143]. Membrane-based isolation techniques are most commonly used for pollen metabarcoding studies, providing a fast and simple way of yielding DNA, although they are costly [142]. Regardless of the technique used, standard principles are followed: first the pollen cell wall (exine) is lysed to enable access to genomic material whilst preventing DNA degradation. Methods for pollen exine rupture can be chemical or mechanical, e.g., bead beating (the most common method) [143]. This lysis step is followed by degradation of the cell membrane, removal of contaminants, and finally precipitation of DNA from protein. Prior to amplification, additional purification steps may be required to remove PCR inhibitors, a common step when using honey as a source of pollen [38].

## 5.3. Amplification

The choice of barcode marker is regarded as one of the most important considerations of DNA barcoding studies and its applications, ultimately affecting the number of taxa recovered and the level of species discrimination obtained [32]. DNA barcode markers require high universality so that a large proportion of species in a sample are amplified, but also low intra-specific and high inter-specific variation for effective species discrimination [33]. Short markers allow for amplification of degraded DNA; however, these come with a caveat of reduced taxonomic resolution [144].

There is no single marker which meets the ideal requirements for a plant barcode [31,32]. For pollen metabarcoding, five regions are commonly used: *rbcL*, ITS2, *matK*, *trnL*, and *trnH-psbA* (Table S1). A multi-locus approach is recommended to ensure the greatest number of taxa are identified [24,38,144]. The length of *matK* (800 bp), restricts its use in metabarcoding due to limitations in read length on standard sequencing platforms [32]. Therefore, it is recommended that *rbcL* and ITS2 are used for pollen metabarcoding, due to their ability to identify taxa at varying taxonomic levels along with additional taxa unique to one marker which provides accurate identification of plant species within mixed pollen samples [38,45,78].

Contamination may also occur in the laboratory; therefore, stringent cleaning procedures are required to minimise these risks. The use of controls (negative in extraction, positive and negative in PCR amplification) helps in the identification of sources of contamination and should be sequenced with samples [38,145]. If sequences occur in negative controls, the number of reads of each taxon should be removed from all samples [81].

## 5.4. Multiplexing and Library Preparation

The ability to scale up metabarcoding studies relies on the use of sample-specific labels in the form of unique sequences of nucleotides which are attached to amplicons. These

unique identifiers allow hundreds or thousands of samples to be pooled for sequencing (multiplexing), significantly increasing the capacity of one sequencing run. Methods for indexing of samples occur either during the initial PCR amplification through nucleotide additions to amplicons or through a secondary PCR amplification along with adapters to allow successful sequencing (library indices) (reviewed in [146]). Each of the methods comes with trade-offs between many factors, mainly the risk of cross-contamination, efficiency of PCR amplification, and overall cost [146]. The two-step PCR approach is most widely used in pollen metabarcoding studies (Table S1), allowing a cost-effective approach to sample labelling whilst allowing effective detection of cross-contamination, but comes with the caveat of increased risk of biases due to an additional amplification stage [146].

### 5.5. Sequencing

Following amplification of DNA, the sequencing strategy used is dependent on a variety of factors including the choice of marker, with most studies thus far utilising the Illumina MiSeq platform. Although concerns are raised over the maximum read length of Illumina platforms ($2 \times 300$ bp) [29,98], multiple studies have demonstrated successful sequencing of longer markers such as *rbcL* (around 500 bp) along with additional adapters and primers [45,80]. Newer sequencing technologies such as the MinION (Oxford Nanopore Technologies) and SMRT platform (PACBIO, Pacific Biosciences) produce longer read lengths, but they generate less reads than Illumina [29]. The development of ultra-deep short read sequencing technologies such as Illumina NovaSeq provide an opportunity to increase sequencing depth and improve the detection rate of taxa. The requirement for high quality and quantity of input DNA may be a limiting factor for some applications of these technologies [49].

### 5.6. Reference Library

The accuracy of DNA barcoding is reliant on a comprehensive reference library [32,147]. The creation of large-scale, complete DNA barcode reference libraries for a national flora has been achieved in the UK [32,139] and Canada [148] using a multi-locus approach, allowing reliable species identification in subsequent pollen metabarcoding studies [37,38]. The curation of reference libraries from chloroplast genomes and nuclear ribosomal DNA sequences can also provide coverage of standard barcodes; however, these methods are more costly [149]. If a complete regional reference database is not available [150], then authors are encouraged to compile custom, relevant reference libraries using the sequences available in GenBank (https://www.ncbi.nlm.nih.gov/genbank/, accessed on 27 January 2022). Curation of these libraries is required, however, to identify and remove incorrect sequences [78,112,116,151]. It is critically important to understand the coverage of the reference library being used compared to the plant taxa that could be detected [32].

### 5.7. Bioinformatic Analysis

The quantity of data produced from DNA metabarcoding studies requires automated processes for curation of sequences, including steps for quality control. The main purpose of this process is to remove any additional nucleotide sequences (index tags, adapter tags, and primers) and to separate each sample for subsequent analysis (demultiplexing). The reduction of the need for expert taxonomists to identify pollen grains is often cited as one of the major advantages of molecular methods over pollen microscopy [91]. However, few authors have highlighted the importance of having good knowledge of the taxonomic group in question (i.e., plants in pollen metabarcoding), including their distribution and phenology for accurate species identification [37,38,83,152]. Misidentifications may occur during the bioinformatic process due to low interspecific variance [32] or incorrectly identified sequences in GenBank [153]. In order to mitigate misidentifications, deployment of a manual verification step in the assignment process, underpinned by botanical expertise, will reduce incorrect species assignments.

*5.8. Towards Standardisation of Methods*

Although each step of the pollen metabarcoding process has a range of different approaches, only certain elements of the entire pollen metabarcoding workflow have been reviewed [132,143,146], leaving a large proportion of the study design to the authors' discretion. Without a standardisation of approaches, comparison of results across multiple studies must be interpreted with caution. Until each stage has been critically reviewed and a robust, standardised approach is established, we encourage researchers to carefully assess the considerations outlined in Table 1 for guidance prior to conducting a pollen metabarcoding study. Further, we call upon authors to be transparent in reporting every aspect of their molecular methods to ensure studies are reproducible, utilising supporting information where word limits are restricting.

*5.9. How Quantitative Is DNA Metabarcoding?*

Finally, there is continued debate over whether DNA metabarcoding may characterise pollen samples in a quantitative manner, with mixed results across studies [74,85,86,154,155]. Quantification has been found to be affected by a combination of marker and primer used, pollen type, mixture characteristics, and PCR conditions [88,156–158]. It is likely that relationships between the proportion of DNA reads and pollen counts are more likely for the most abundant taxa within a sample [83,159]. Similar to microscopy, rare taxa are difficult to detect using pollen metabarcoding [87]. Whilst this is a limitation, studies examining insect floral resource use often place greater focus on those plants detected at higher abundance. For this reason, along with the potential biases which can occur, DNA metabarcoding should be considered as semi-quantitative and relative read abundance used for downstream analysis [160]. We do not recommend the use of presence/absence approaches due to rare taxa being overstated and abundant taxa devalued [160].

## 6. Opportunities and Future Directions

The use of DNA metabarcoding as a tool to investigate pollinator foraging has allowed increased insight into the interactions between plants and pollinators; however, it is still a developing field. Most studies focus on the identification of pollen; however, other plant material may be used to identify relationships between insects and plants. For example, recently, the characterisation of resin within the nests of solitary bees through DNA metabarcoding has been suggested as a promising approach to identify which plants are important for nest building [161]. DNA metabarcoding is also not free from limitations. Overall, the greatest limitation is the cost and reproducibility of the molecular techniques [162], which determine which methods are used. Whilst the interpretation of data remains semi-quantitative, future work may lead to the ability to accurately measure pollen abundance, significantly improving the application of this technique [157,158]. Quantification may be improved by using PCR-free approaches which also provide a greater representation of the genome [46]. Recent work by Bell et al. [46] has demonstrated that whole-genome shotgun sequencing of pollen DNA is a reliable method for identification of pollen species mixtures. However, coverage of eukaryotic organisms in reference libraries remains low, and assembly of whole genomes is currently more expensive than metabarcoding per sample [46]. It is likely that DNA metabarcoding will remain the standard technique until genome-level coverage improves. Until then, genome-skimming techniques may hold promise to identify beyond the species level, e.g., to population or individual, if the nuclear genome is retained [163].

## 7. Final Remarks

This review describes the range of approaches available to investigate floral visitation by pollinators using DNA metabarcoding. We demonstrate how the ability to yield valuable individual-to-community-level information on foraging over large spatiotemporal scales allows for a breadth of ecological questions to be explored, for the benefit of both the conservation of pollinators and the maintenance of the ecosystem services they provide.

DNA metabarcoding has become a standard tool for the characterisation of complex plant–pollinator interactions, allowing for an improved understanding of threatened global biodiversity.

**Supplementary Materials:** The following supporting information can be downloaded at https://www.mdpi.com/article/10.3390/d14040236/s1, Table S1: Details of studies which use plant DNA metabarcoding to identify floral visitation by pollinators or developed methods to support. Table S2: Recommended primer sequences used to amplify the *rbcL* and ITS2 barcode regions.

**Author Contributions:** Conceptualisation, A.L., N.d.V. and S.C.; methodology, A.L. and N.d.V.; validation, A.L., L.J., L.W., S.C. and N.d.V.; investigation, A.L.; data curation, A.L.; writing—original draft preparation, A.L. and N.d.V.; writing—review and editing, A.L., L.J., L.W., S.C. and N.d.V.; supervision, N.d.V. and S.C.; funding acquisition, N.d.V. All authors have read and agreed to the published version of the manuscript.

**Funding:** A.L., L.J. and N.d.V. have received funding through the Welsh Government Rural Communities—Rural Development Programme 2014–2020, which is funded by the European Agricultural Fund for Rural Development and the Welsh Government. A.L. and L.W. were supported by a Knowledge Economy Skills Scholarship (KESS2), part funded by the Welsh Government's European Social Fund (ESF).

**Institutional Review Board Statement:** Not applicable.

**Data Availability Statement:** All relevant data are provided in the Supplementary Materials.

**Conflicts of Interest:** The authors declare no conflict of interest.

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
