# Peer review of "Using DNA Metabarcoding to Identify Floral Visitation by Pollinators"

_diversity, doi:10.3390/d14040236_

Round 1

Reviewer 1 Report

In the manuscript “Using DNA Metabarcoding to Identify Floral Visitation by Pollinators” Lowe et al. provide a review of how metabarcoding methods are being used to identify flower visits by pollinators. The manuscript is well written and includes many relevant perspectives and recent developments in the literature. My main concern is that the review is unfocused and not well balanced in how it compares metabarcoding to other methods for identifying flower visits. Methods such as waggle dance analysis and harmonic radar are only mentioned briefly in the text so a full comparison of these methods to metabarcoding in Table 1 comes off as unjustified. A number of statements about observations of visits also appear quite one-sided and unjustified. If the authors want to compare all these methods, the review needs a broader topic and they need to thoroughly research the other methods. This includes reflections on which questions such methods might be able to answer that metabarcoding can’t. Most likely they will find that all methods have value but some are more appropriate than others under different circumstances. The manuscript needs a better balance in how it compares meta-barcoding and other methods or else it comes off as opinions rather than a review. Alternatively, the authors should focus on metabarcoding and leave out these comparisons.

67-69: It is not clear how preference for plant material (such as leaves) for solitary bee nests is relevant for the review which focuses on flower visits in terms of pollination. I would suggest removing this entirely or explain its relevance. This comes up again towards the end of the review (lines 413-414).

88-89: I don’t know what the authors mean by the statement that “plant-pollinator interactions and networks to be identified and constructed quickly and cheaply”– considering also the following sentences that describe problems with sampling bias in plant-pollinator networks. Networks are notoriously labour intensive to sample to even a low degree of completeness (see for example Chacoff et al. 2012). If it’s done quickly and cheaply it is most likely not done correctly. One of the benefits from using alternative methods, such as metabarcoding, is that you might save some labour costs and get more data for your money.

94-96: It is true that observing plants mean that the data will lack the insect perspective. Doing this the observer is also likely to miss visits by rare insects. However, doing it the other way around – observing insects – is also likely to result in bias. Presenting both sides of this makes the case for using metabarcoding stronger in that it is the insect perspective but with higher likelihood of observing rare plant visits.

118-123: this is partly correct but many who do these studies catch the pollinators individually and identify them later in the laboratory. This is for example standard for many solitary bees that cannot be identified in the field – even by experts. This is no different from the morphological identification that might also be used for the pollinators when using metabarcoding of pollen (as written in line 121). This claim should also be revised in Table 1.

Line 154-211: Most of the examples provided here are from honeybee studies. More examples from wild pollinators would be more compelling. Does it reflect the literature that most are on honeybees? If so, can the authors reflect on why there aren’t more studies of wild pollinators under these topics. Is there some barrier to the uptake of the method in less controlled systems?

303-304: there seems to be a word missing. Should it be “although they are costly”?

Table 1: I don’t think the authors have provided support for the advantages/disadvantages of several methods they are summarising in this table (observations, waggle dance and harmonic radar). The review is about metabarcoding methods to identify interactions and many of these methods are clearly outside of the scope of the review. It also comes across have not evaluated these methods fully against each other. For example, the authors are making quite a few statements about observing visits (first row) that are unjustified or at least very simplified. The claim that observing visits/interactions is cheap is incorrect if you consider labour costs. I also don’t see an explanation for the claim that these studies often result in qualitative data only. One of the benefits of observing interactions is that you can, for example, record the number of individuals visiting a flower. This is used as interaction frequency in ecological networks. If the authors want to challenge those methods for obtaining quantitative data they should do that in the text. But then the topic of the review should also be broadened.

Table 1: first row - “observing plants or pollinators” is a bit confusing – if you are observing visits you are observing both (but of course with either a plant or pollinator focus)?

Chacoff, N. P., Vázquez, D. P., Lomáscolo, S. B., Stevani, E. L., Dorado, J., & Padrón, B. (2012). Evaluating sampling completeness in a desert plant–pollinator network. Journal of Animal Ecology, 81, 190–200. https://doi.org/10.1111/j.1365-2656.2011.01883.x

Author Response

Please see the attachment for a response to all reviewers' comments. 

Reviewer 2 Report

Dear Authors, 

the topic of this manuscript is very interesting and innovative. I think it is well organised and well written. I suggest that the tables be modified and rationalised because they are too long and not very readable. In table 1 in the sections "pollen microscopy", "pollen metabarcoding" and "metagenomics" there are many common advantages and disadvantages that could be somehow grouped together. Furthermore, on page 7 in paragraph 4.4 the scientific names of the plants should be written in italics. 

Author Response

Please see attached for a response to all the reviewers' comments.

Reviewer 3 Report

The authors of this review present information for usage of DNA metabarcoding to identify various pollinators.

I have some suggestions to improve the paper.

Line 29. Please, replace function with sustainability.

Line 34. Please, comment the following papers:

Neov B, Georgieva A, Shumkova R, Radoslavov G, Hristov P. Biotic and Abiotic Factors Associated with Colonies Mortalities of Managed Honey Bee (Apis mellifera). Diversity. 2019; 11(12):237. https://doi.org/10.3390/d11120237

Hristov P, Shumkova R, Palova N, Neov B. Factors Associated with Honey Bee Colony Losses: A Mini-Review. Veterinary Sciences. 2020; 7(4):166. https://doi.org/10.3390/vetsci7040166

Goulson, D.; Nicholls, E.; Botías, C.; Rotheray, E.L. Bee declines driven by combined stress from parasites, pesticides, and lack of flowers. Science 2015, 347, 1255957.

Line 64. environmental sample using DNA – put in bracket [eDNA].

In Table 1, the authors can divide the methods as molecular and non-molecular.

Line 122. CO1 – Please, replace with co1.

Line 166. By sequencing pollen - the pollen cannot be sequenced, rather the eDNA contained in the pollen.

Line 238. Bee bread is a slightly different composition from pollen. Bee bread is known to have a high concentration of protein and other nutritional impacts making it more valuable to natural food nutritionists.

Line 250-251. European goldenrod Solidago virgaurea, Leucanthemum vulgare and yarrow Achillea millefolium – species name should be in italic.

Line 316. rbcL and matK, with trnH-psbA – Please, give a full name of these genes and after that put abbreviations in brackets. Also, change in italic the gene names.

Line 328. PCR – replace with PCR amplification. Also in Lines -336, 337 and 340.

Line 364. Please, add this link after GenBank https://www.ncbi.nlm.nih.gov/genbank/.

The authors did a great job. Congratulations!

The reference list does not follow the requirements from Diversity journal.

Author Response

(The authors gave the same response as above.)
